# Study of Hydraulic Disturbance Transient Processes in Pumped-Storage Power Stations Considering Electro-Mechanical Coupling

**DOI:** 10.3390/s26010311

**Published:** 2026-01-03

**Authors:** Chengpeng Liu, Zhigao Zhao, Xiuxing Yin, Jiandong Yang

**Affiliations:** State Key Laboratory of Water Resources Engineering and Management, Wuhan University, Wuhan 430072, China

**Keywords:** pumped-storage power station, load rejection, hydraulic disturbance, frequency regulation, numerical simulation

## Abstract

Pumped-storage power stations, as a critical resource for supporting secure and stable grid operation, typically adopt a ’single-tunnel-multiple-unit’ configuration, where hydraulic disturbance becomes a key operating condition affecting system security. Existing studies have primarily focused on the impact of the hydro-mechanical subsystem on the normally operating units, while the influence of the electrical subsystem on hydraulic disturbance has been insufficiently addressed. To bridge this gap, this study develops a coupled model of a grid-connected pumped-storage power station incorporating a detailed representation of the power system. The model comprehensively captures the multi-domain interactions among the hydraulic, mechanical, electrical, and grid subsystems, and its accuracy is validated using data from a physical model test platform. On this basis, the hydraulic transient responses under two modeling conditions—detailed grid representation and conventional simplified grid modeling—are systematically compared. Key parameters from the hydraulic, mechanical, and electrical domains are further examined to quantify their impacts on the dynamic characteristics of hydraulic disturbance. The results demonstrate that detailed grid modeling reveals novel characteristics of the hydraulic disturbance that cannot be simulated by the conventional model. Under the detailed model, the normally operating units compensate for the power deficit caused by the tripping unit, leading to reduced hydraulic pressure fluctuations and a significant increase in the maximum output of the operating units. Meanwhile, hydro-mechanical parameters strongly influence the transient behaviors of unit output and net head, whereas the guide vane regulation of the operating unit remains predominantly driven by grid-frequency deviations. Overall, this study enhances the understanding of hydraulic disturbance dynamics in grid-connected pumped-storage systems and provides important insights for ensuring their secure and stable operation.

## 1. Introduction

With the increasing penetration of renewable energy sources in modern power grids, maintaining system stability has become considerably more challenging, placing higher demands on renewable energy accommodation and regulation capabilities [1]. As the most effective large-scale energy storage technology currently available [2], pumped-storage power stations have become an essential resource for supporting secure and stable grid operation, and their deployment is accelerating worldwide [3]. According to statistics from the National Energy Administration of China, the cumulative installed capacity of pumped-storage power stations in China exceeded 58 GW by the end of 2024 [4].

Unlike conventional hydropower stations, which typically adopt a “one-tunnel-one-unit” layout, pumped-storage power stations commonly employ a “single-tunnel-multiple-unit” configuration. Owing to the strong hydraulic coupling within the water conveyance system, pressure fluctuations can propagate between units through the bifurcated tunnel system [5]. When a load rejection occurs in one of the units within the same hydraulic branch, the resulting water-hammer pressure is transmitted through the conduit to adjacent units that remain in operation, causing significant alterations in their dynamic behavior. Such operating conditions are generally referred to as hydraulic disturbance conditions [6]. Under these circumstances, severe pressure oscillations may induce large fluctuations in unit output, potentially exceeding the generator’s safety margin and endangering stable unit operation [7]. Therefore, hydraulic disturbance represents a critical operating condition in pumped-storage power stations and warrants careful attention.

Pumped-storage power stations represent typical multi-physics, strongly coupled nonlinear systems [8], and elucidating the dynamic mechanisms of hydraulic disturbance transients remains a core challenge in related research. Although physical model tests and prototype measurements provide the most direct means for observing such processes [9], their applicability is considerably limited by constraints on sensor placement, the complexity of constructing experimental platforms, and operational safety requirements of actual stations [10]. As a result, numerical simulation has become the primary approach for investigating hydraulic disturbance conditions [11]. In numerical modeling, a pumped-storage power station must be represented comprehensively from the hydraulic, mechanical, and electrical perspectives. On the hydraulic side, Zhang [12] applied the rigid water-hammer model to simulate the water conveyance system; however, this approach is more suitable for small-disturbance scenarios. In contrast, the method of characteristics (MOC), owing to its ability to handle complex boundary conditions, has been widely adopted for hydraulic modeling in pumped-storage systems [13]. The mechanical subsystem comprises the pump-turbine and the governor system. The latter is often described using a PI controller combined with an actuator, resulting in a relatively unified and simple structure; and Lai [14] and Tan [15] developed mathematical models of the governor under speed-control and power-control modes, respectively. The pump-turbine itself, however, features complex internal mechanisms and diverse physical behaviors, and no universally accurate model is yet available [16]. Consequently, its dynamic characteristics are typically represented using complete characteristic curves [17,18], with dynamic solutions obtained through spatial surface interpolation [18] or the Suter transformation [19]. On the electrical side, Zhao [20] developed a detailed generator model, but substantial simplifications were made in the turbine and water conveyance components, limiting its capability to reflect full-condition dynamics. Zhang [21] modeled grid behavior using an equivalent generator representation or constructed a more comprehensive transient power system model [22]. However, many studies either neglect the water conveyance system of pumped-storage stations [23,24] or model it using a simplified “one-tunnel-one-unit” rigid water-hammer representation [25], which deviates from the practical “single-tunnel-multiple-unit” configuration of pumped-storage power stations. These limitations highlight the existing gap between current research models and real-world engineering applications.

A number of studies have been conducted to investigate the dynamic characteristics of hydraulic disturbance. Rezghi [26] developed a simulation model of a pumped-storage power station with a single-tunnel-two-unit configuration and found that adopting a speed-control mode in the governor of the operating unit can effectively improve the hydraulic transient performance. Guo [27] analyzed hydraulic disturbance in a long water conveyance system with a one-tunnel-two-unit layout and examined the interaction characteristics between different units. Cao [28] investigated the dynamic behaviors of fixed- and variable-speed pumped-storage units under various water conveyance configurations, and conducted comparative analyses of the factors influencing the transient responses of the operating units. Cui [29,30] examined the hydraulic superposition characteristics in a single-tunnel-two-unit pumped-storage system under successive load rejections in the hydraulic short-circuit mode, revealing the underlying mechanism of water-hammer pressure superposition.

Most existing studies on hydraulic disturbance in pumped-storage power stations have predominantly focused on the hydro-mechanical domain, examining the influence of hydraulic and mechanical parameters on the transient responses of normally operating units. However, these studies usually involve substantial simplifications of the electrical subsystem, typically adopting single-machine isolated-grid models or infinite-bus representations. Such modeling approaches result in an overly simplified description of the power system, especially in the context of modern power system structures with increasing complexity and high penetration of renewable energy sources.As a result, these simplified models are generally unable to capture the electrical coupling effects between the power grid and the generating units, thereby limiting their ability to reflect the coupled dynamic characteristics of the pumped storage units under realistic operating conditions. To address this issue, the present study reexamines the dynamic characteristics of hydraulic disturbance transients based on a refined and detailed power system model. The main contributions of this work are as follows: (1) A mathematical model of the transient processes in pumped-storage power stations is developed, incorporating a detailed representation of grid dynamics that captures variations in both voltage and frequency, as well as the dynamic characteristics of individual system components. The accuracy of the proposed model is validated using data obtained from a physical model test platform. (2) The hydraulic disturbance transient processes under detailed grid modeling are investigated, and their differences and underlying mechanisms are compared against those obtained using conventional simplified grid models. (3) Key parameters from the hydraulic, mechanical, and electrical domains are analyzed to determine their influence on the dynamic characteristics of hydraulic disturbance, and the corresponding physical mechanisms are discussed.

The remainder of this paper is organized as follows. Section 2 presents the mathematical models and simulation methods for the pumped-storage power station and the associated power system. Section 3 validates the adopted models using data from a physical model test platform. Section 4 compares the dynamic responses of hydraulic disturbance transients with and without detailed grid modeling, and investigates the mechanisms by which various factors influence hydraulic disturbance. Section 5 discusses aspects that warrant further exploration in future work. Finally, Section 6 concludes the paper.

## 2. Mathematical Modeling

When the Electro-Mechanical coupling is taken into account, a grid-connected pumped-storage power station can be regarded as a composite system consisting of the water conveyance system, the governor system, the hydro-turbine generator unit, and the electrical power system. This section provides an overview of these subsystems.

### 2.1. Model of Hydraulic Conduit System

Without considering pipe elasticity and water compressibility, the mathematical model of a pressurized water conveyance system is typically formulated using the continuity equation and the momentum equation, which can be expressed as follows [31]: (1)$$V\frac{\partial H}{\partial x}+\frac{\partial H}{\partial t}+\frac{a^{2}}{g}\frac{\partial V}{\partial x}+\frac{a^{2}}{gA}\frac{\partial A}{\partial x}V-V\sin\alpha =0$$(2)$$g\frac{\partial H}{\partial x}+V\frac{\partial V}{\partial x}+\frac{\partial V}{\partial t}+\frac{fV\left|V\right|}{2D}=0$$
where *H* is the pressure head; *V* is the flow velocity; *x* is the axial coordinate along the pipeline; *D* is the pipe diameter; *g* is the gravitational acceleration; *A* is the cross-sectional area of the pipeline; α is the pipe slope; *f* is the Darcy-Weisbach friction coefficient.

In general, hyperbolic partial differential equations do not admit closed-form analytical solutions. However, they can be transformed into ordinary differential equations along two characteristic lines using the method of characteristics, and subsequently discretized to obtain the flow variables at each computational node. This procedure ultimately leads to a set of linear algebraic equations, as expressed in: (3)$$\left\{\begin{array}{c} C^{+}:Q_{P}^{t+1}=Q_{CP}-C_{QP}H_{P}^{t+1}\\ C^{-}:Q_{P}^{t+1}=Q_{CM}+C_{QM}H_{P}^{t+1} \end{array}\right.$$
where the expressions of the parameters are given in Equation (Equation 4). The subscript P denotes the current grid point, while the subscripts R and S represent the adjacent upstream and downstream nodes, respectively.(4)CQP=1C−C3C−C3APAP+CC1+C2CQM=1C+C3C+C3AP+C(C4+C5)AP+C(C4+C5)QCP=CQP[QR(Ct+Ct3AP−CtCt1)+HR]QCM=CQM[QS(Ct−Ct3AP−CtCt4)−HS]Ct=ag,Ct1=a(AP−AR)2AP(aAR+QP)Ct2=ΔtSPQR8ARAP2f,Ct3=12ΔtsinαCt4=a(AP−AS)2AP(aAS−QS),Ct5=ΔtSPQS8ASAP2f
where $C_{QP}$, $C_{QM}$, $Q_{CP}$, and $Q_{CM}$ are intermediate variables calculated from Equation (Equation 3). The coefficients $C_{t}$, $C_{t1}$, $C_{t2}$, $C_{t3}$, $C_{t4}$, and $C_{t5}$ are auxiliary intermediate variables introduced for the computation of $C_{QP}$, $C_{QM}$, $Q_{CP}$, and $Q_{CM}$.

For a multi-unit pumped-storage system, the water conveyance system includes not only pressurized pipelines but also boundary components such as the upstream and downstream reservoirs, bifurcated tunnels, and surge tanks. Due to space limitations, this paper introduces only the mathematical model of the surge-tank boundary, while the remaining boundary conditions can be found in the referenced literature [32,33].

### 2.2. Model of Pumped-Storage Unit and Generator

The hydro-turbine generating unit is the core component in a pumped-storage power station responsible for the energy conversion among hydraulic, mechanical, and electrical domains. It typically consists of a turbine and a generator, which are connected through a shaft system. The turbine is a highly nonlinear device, and no universally accurate mathematical model exists to fully describe its transient behavior. Therefore, in hydraulic-electrical transient analyses of hydropower stations, the dynamic characteristics of the pump-turbine, such as guide vane opening (GVO), flow rate, rotational speed, head, and torque, are generally represented using the complete characteristic curves.

The complete characteristic curves are established based on the steady-state measurements of model runner operating points and then extended to cover the full operating range. To reduce discrepancies between the model runner and the prototype runner, unit parameters are commonly employed to describe the turbine speed, discharge, and torque characteristics. The conversion relationships between the unit parameters and the actual runner parameters are given as follows [13]: (5)$$Q_{11}=\frac{Q}{D_{1}^{2}\sqrt{H}};N_{11}=\frac{ND_{1}^{2}}{\sqrt{H}};M_{11}=\frac{M}{D_{1}^{3}\sqrt{H}}$$
where *Q* is the discharge, $m^{3}/s$; *N* is the rotational speed, r/min; *M* is the turbine torque, N·m; *H* is the pump-turbine working head, m; $D_{1}$ is the runner diameter, m; $Q_{11}$ is the unit discharge, $m^{3}/s$; $N_{11}$ is the unit speed, r/min; and $M_{11}$ is the unit torque, N·m. Based on the unit parameters, the mathematical model of the complete characteristic curves for the pump-turbine can be expressed as: (6)$$\left\{\begin{array}{c} Q_{11}=f\left(y,N,H\right)\\ M_{11}=g\left(y,N,H\right) \end{array}\right.$$
where *y* denotes the GVO of the turbine; *f* represents the discharge characteristic curve; and *g* represents the torque characteristic curve.

In conventional transient calculations, the generator’s electrical transients are typically neglected. As a result, the dynamic behavior of the hydro-turbine generating unit is simplified to the rotor motion equation, which can be expressed as follows [34]: (7)$$J\frac{d\omega_{r}}{dt}=T_{m}-T_{em}$$
where *J* is the moment of inertia of the hydro-turbine generator unit; $\omega_{r}$ is the mechanical angular speed of the rotor; $T_{m}$ is the turbine mechanical torque; and $T_{em}$ is the electromagnetic torque.

When a disturbance occurs in the power system, the dynamic behavior of the hydro-turbine generator unit is affected, making it necessary to account for the generator’s electrical transients. Considering the required accuracy and computational efficiency of electromechanical transient simulations, the subtransient model with an equivalent rotor circuit is adopted for salient-pole synchronous generators in accordance with IEEE standards. The fundamental model of the salient-pole synchronous generator is given as follows [20]: (8)$$\left\{\begin{array}{c} {T^{\prime }}_{d0}\frac{d{E^{\prime }}_{q}}{dt}=E_{f}-\frac{X_{d}-X_{\sigma }}{{X^{\prime }}_{d}-X_{\sigma }}{E^{\prime }}_{q}+\frac{X_{d}-{X^{\prime }}_{d}}{{X^{\prime }}_{d}-X_{\sigma }}{E^{\prime \prime }}_{q}-\frac{\left(X_{d}-{X^{\prime }}_{d}\right)\left({X^{\prime \prime }}_{d}-X_{\sigma }\right)}{{X^{\prime }}_{d}-X_{\sigma }}i_{d}\\ {T^{\prime \prime }}_{d0}\frac{d{E^{\prime \prime }}_{q}}{dt}={T^{\prime \prime }}_{d0}\frac{{X^{\prime \prime }}_{d}-X_{\sigma }}{{X^{\prime }}_{d}-X_{\sigma }}\frac{d{E^{\prime }}_{q}}{dt}-{E^{\prime \prime }}_{q}+{E^{\prime }}_{q}-\left({X^{\prime }}_{d}-{X^{\prime \prime }}_{d}\right)i_{d}\\ {T^{\prime \prime }}_{q0}\frac{d{E^{\prime \prime }}_{d}}{dt}=-{E^{\prime \prime }}_{d}+\left(X_{q}-{X^{\prime \prime }}_{q}\right)i_{q}\\ \frac{d\delta }{dt}=\omega_{r}-\omega_{s} \end{array}\right.$$
where $E_{q}^{\prime }$ is the *q*-axis transient electromotive force; $E_{q}^{\prime \prime }$ is the *q*-axis subtransient electromotive force; $E_{d}^{\prime \prime }$ is the *d*-axis subtransient electromotive force; $i_{d}$ and $i_{q}$ are the dq-axis stator current; $X_{d}$ and $X_{q}$ are the dq-axis synchronous reactance; $X_{d}^{\prime }$ is the *d*-axis transient reactance; $X_{d}^{\prime \prime }$ and $X_{q}^{\prime \prime }$ are the dq-axis subtransient reactance; $X_{\sigma }$ is the stator leakage reactance; $T_{d0}^{\prime }$ is the *d*-axis transient time constant; $T_{d0}^{\prime \prime }$ and $T_{q0}^{\prime \prime }$ are the dq-axis subtransient time constant; δ is the generator power angle; $\omega_{r}$ is the rotor angular frequency; and $\omega_{s}$ is the synchronous angular frequency.

The relationship between the transient electromotive force and the stator terminal voltage can be expressed as follows: (9)$$\left\{\begin{array}{c} u_{d}={E^{\prime \prime }}_{d}+{X^{\prime \prime }}_{q}i_{q}-R_{s}i_{d}\\ u_{q}={E^{\prime \prime }}_{q}-{X^{\prime \prime }}_{d}i_{d}-R_{s}i_{q} \end{array}\right.$$
where $u_{d}$ and $u_{q}$ are the dq-axis stator terminal voltages of the generator, $R_{s}$ is the stator resistance. At this point, the electromagnetic torque can be expressed as: (10)$$T_{em}=E_{q}^{\prime \prime }i_{q}+E_{d}^{\prime \prime }i_{d}-\left(X_{d}^{\prime \prime }-X_{q}^{\prime \prime }\right)i_{d}i_{q}$$

The excitation system supplies the generator with excitation current and regulates this current to control the generator terminal voltage, ensuring that it remains at its rated level and satisfies power quality requirements. The corresponding mathematical models can be found in the referenced literature [35].

As the key device governing the guide vane movement of pumped-storage units, the governor helps maintain system frequency and enhances overall system stability. It is typically composed of a PI controller and an actuator [19] and its mathematical model can be expressed as follows [36]: (11)$$\left\{\begin{array}{c} T_{y}\frac{d\Delta y}{dt}+\Delta y=-\left[K_{P}\left(\omega_{rref}-\omega_{r}+b_{p}\Delta y\right)+K_{I}x_{g}\right]\\ \frac{dx_{g}}{dt}=\omega_{rref}-\omega_{r}+b_{p}\Delta y \end{array}\right.$$
where $K_{P}$ is the proportional gain of the governor; $K_{I}$ is the integral gain; $\omega_{rref}$ is the reference speed; $T_{y}$ is the inertia time constant; $x_{g}$ is the intermediate variable; and $b_{p}$ is the steady-state parameter.

### 2.3. Model of the Power System

The mathematical model of power system transient stability simulation in the time domain consists of a set of stiff nonlinear differential-algebraic equations. The general solution approach is to apply numerical integration to convert the differential equations into discrete difference equations, which are then solved together with the algebraic equations as a system of nonlinear algebraic equations. The differential-algebraic formulation for power system transient stability analysis can be written as [37]: (12)$$\left\{\begin{array}{c} \dot{X}=F\left(X,V\right)\\ YV=I\left(X,V\right) \end{array}\right.$$
where *X* and *V* denote the state variables and algebraic variables, respectively; *Y* is the network admittance matrix, which is determined by the system topology and line parameters and changes only during faults or switching operations; *F* is the vector function defined by the mathematical models of the dynamic components; and *I* is the current injection vector determined by the interface equations between the dynamic components and the network. The implicit trapezoidal integration method is widely used in large-scale commercial power system analysis programs. After discretization using the implicit trapezoidal method, the differential-algebraic equations can be transformed into [38]: (13)$$\left\{\begin{array}{c} X_{n+1}=X_{n}+0.5h\left[F\left(X_{n},V_{n}\right)+F\left(X_{n+1},V_{n+1}\right)\right]\\ YV_{n+1}=I\left(X_{n+1},V_{n+1}\right) \end{array}\right.\;$$
where *n* denotes the integration time step of the simulation, and *h* represents the time-step size. In addition to the pumped-storage power station, the power system also includes network topology, thermal power units, loads, and other components. The corresponding mathematical models for these elements can be found in the referenced literature [35].

Since the grid frequency is primarily maintained through the synchronous characteristics of synchronous generators, the center-of-inertia frequency is used to represent the system frequency variation. The relationship between the grid frequency and the inertia of individual synchronous generators can be expressed as follows [39]: (14)$$f_{COI}=\frac{\sum H_{i}S_{i}f_{i}}{H_{sys}S_{sys}}=\frac{\sum H_{i}S_{i}f_{i}}{\sum H_{i}S_{i}}$$
where $f_{COI}$ is the center-of-inertia (COI) frequency; $H_{i}$, $S_{i}$, $f_{i}$ are the inertia constant, rated capacity, and frequency of the *i*-th generator, respectively; $H_{sys}$ is the equivalent system inertia; and $S_{sys}$ is the rated capacity of the grid. In essence, the grid frequency can be regarded as the weighted average of the frequencies of all synchronous generators.

## 3. Experimental Testing and Model Validation

### 3.1. Experimental Platform of Pumped Storage System

To validate the accuracy of the proposed model, load-rejection experiments were conducted on a variable-speed pumped-storage model test platform. The platform consists of several subsystems, including the pipeline system, circulating water channel, variable-speed pump-turbine, doubly fed induction motor, governor system, AC excitation and electrical protection system, load system, coordinated controller, monitoring system, pressure-regulation water system, and measurement system. The water conveyance system adopts a ‘one-tunnel-two-unit’ configuration. The upstream conduit is connected to a sealed vacuum tank equipped with an air-release valve, and the upstream water level is adjusted by a variable-frequency pump regulating the water level in the vacuum tank. The downstream conduit is connected to an open-type downstream reservoir. The circulating water subsystem provides stable, wide-range, and continuous water flow for the experiment while effectively reducing water consumption. The overall schematic diagram and on-site layout of the test platform are shown in Figure 1.

As illustrated in Figure 1, the platform is equipped with multiple sensors and data acquisition devices for real-time measurement and recording throughout the entire experiment. Pressure sensors are installed at key locations, including the upstream reservoir, downstream reservoir, surge tank, volute inlet, and draft-tube outlet. Dynamic level gauges are used at the volute inlet and draft-tube outlet, while miniature pressure sensors are deployed at the remaining points. Mechanical rotational speed is monitored using both the gear-disk frequency method and the residual-voltage frequency method, with cross-validation to ensure measurement accuracy. The GVO is measured by an angular displacement sensor, and electrical parameters are directly obtained from the control cabinet. The data acquisition system consists of 32 channels, allowing simultaneous access to various types of sensors and enabling synchronized multi-physical measurements and analysis.

### 3.2. Model Verification and Validation

At the initial moment, the upstream reservoir level is set to 36.88 m and the downstream reservoir level to 8.14 m. The pump-turbine operates at 0.75 pu, and since the unit belongs to a variable-speed pumped-storage system, its initial rotational speed is set to 920 r/min. Under these conditions, the flow rate through the unit is 273.4 L/s and the power output is 66 kW. At 24.5 s, the unit undergoes a sudden load rejection, after which the governor regulates the rotational speed back to its rated value. To ensure consistency with the dynamic behavior observed in the experiment, the GVO trajectory measured during the test is directly applied to the numerical simulation. The corresponding computational results are shown in Figure 2.

As shown in Figure 2, the simulation method employed in this study can accurately predict the hydraulic transient characteristics of the load-rejected unit. After the load-rejection command is issued on the physical model test platform, the unit speed rises rapidly once the load is removed. To prevent excessive overspeed, the governor responds immediately, causing a rapid closure of the GVO. This leads to a sharp reduction in flow rate, an increase in volute pressure, and a decrease in draft-tube pressure. The simulation results agree well with the experimental measurements in both dynamic trends and key characteristic values. The prediction error of the maximum volute pressure is 1.59 m, corresponding to a relative deviation of 3.03%. The error in the minimum draft-tube pressure is 0.22 m, with a relative deviation of 3.71%. In addition, the deviation in the peak rotational speed is 31.67 r/min, yielding a relative deviation of 2.59%. It should be noted that slight oscillations appear in the initial stage, as shown in Figure 2c,d, mainly due to theoretical simplifications when transitioning from steady-state to transient calculations. In addition, the electromagnetic flowmeter used in the experiment cannot measure reverse flow; therefore, when flow reversal occurs, the flowmeter output drops to zero. Moreover, discrepancies between the characteristic curves used in the simulation and the actual characteristics of the physical model, along with unavoidable differences in certain structural and parametric details, may also introduce deviations.

Overall, the simulation model and method developed in this study can accurately reproduce the hydraulic transient behavior of the unit under load-rejection conditions. Its results show good agreement with the physical experiments, demonstrating that the model is suitable for analyzing the dynamic characteristics and underlying mechanisms of pumped-storage units under hydraulic disturbance conditions.

## 4. Analysis of Hydraulic Disturbance Transient Processes

To investigate the impact of grid connection on the hydraulic disturbance behavior of pumped-storage power stations, it is necessary to construct a coupled simulation model that integrates both the power system and the pumped-storage units. Therefore, a pumped-storage power station with a “single-tunnel-two-unit” configuration is embedded into the IEEE 39-bus power system for coupled dynamic simulation, as illustrated in Figure 3. In this system, the two units of the pumped-storage station are connected to Buses 30 and 37, respectively. The main parameter settings of the power system can be found in [40].

### 4.1. Comparison of the Electrically Coupled Model and the Traditional Model

To further investigate the differences between hydraulic disturbance computed with a detailed power-system model and that obtained using the traditional simplified-grid approach, hydraulic disturbance transient simulations were carried out under different control modes. Under the rated head, both units operate at rated load and are connected to the grid under normal conditions. A sudden 100% load rejection is applied to Unit 1#. The corresponding dynamic responses of Unit 2# under various control modes, as well as the differences from the traditional hydraulic disturbance calculation method, are shown in Figure 4.

As shown in Figure 4, the hydraulic transient responses of the operating unit exhibit significant differences between the detailed grid model and the conventional simplified grid model. Traditional analyses of hydraulic disturbance typically assume that the two units of a pumped-storage power station are connected to two independent grids. Under this assumption, the tripped unit exerts no electrical influence on the operating unit, and the operating unit is affected solely by the water-hammer pressure induced by the load rejection. However, when detailed grid dynamics are included, in addition to the conventional hydraulic effects, electrical-domain coupling introduces a substantial impact on the response of the operating unit.

For pumped-storage units, the pump-turbine is electrically coupled to the grid through a synchronous generator, meaning that the unit speed is directly tied to grid frequency. When a load rejection occurs in one of the units, the grid frequency drops, and the speed of the operating unit decreases correspondingly. In this case study, the operating unit speed falls to 0.96 pu under the detailed grid model. To maintain speed stability, the governor increases the GVO. In contrast, under the simplified grid model, the two units are electrically independent, and the tripped unit influences the operating unit only through hydraulic transmission. The rapid closure of the GVO of the tripped unit leads to an increase in volute pressure and a decrease in draft-tube pressure, thereby increasing the net head and driving torque of the operating unit. As a result, the unit speed rises; to maintain speed stability, the governor reduces the GVO. In this process, the operating unit speed increases to 1.067 pu and the gate opening decreases to 0.587 pu under the simplified grid model.

The change in GVO directly affects the internal pressure distribution of the unit. A decrease in gate opening increases volute pressure; when this combines with the pressure rise caused by the tripped unit, the simplified grid model yields a higher peak volute pressure. Conversely, an increase in gate opening reduces volute pressure, partially offsetting the water-hammer effect induced by the tripped unit, resulting in a lower volute pressure under the detailed grid model. A similar trend appears in the draft-tube pressure: the simplified model produces lower draft-tube pressure, while the detailed model produces higher values, as shown in Figure 4b,c. In this scenario, the detailed grid model yields a maximum volute pressure of 483.72 m and a minimum draft-tube pressure of 38.18 m, whereas the simplified model yields 474.78 m and 39.14 m, respectively.

Furthermore, because the simplified grid model neglects electrical power coupling, the output of the operating unit oscillates due to water-hammer effects but eventually returns to its initial steady-state value. In contrast, when the detailed grid model is considered, the operating unit must compensate for the power deficit caused by the load rejection. Its power output gradually increases, superimposed on the transient oscillations driven by water-hammer pressure. Limited by the maximum gate opening, the unit eventually stabilizes at the maximum power output corresponding to the available head, as shown in Figure 4f. In this case, the maximum power output is 413.74 MW under the detailed model, compared with 357.9 MW under the simplified model. Overall, detailed grid modeling fundamentally alters the dynamic response characteristics of pumped-storage units under hydraulic disturbance conditions and highlights the critical role of electrical coupling in hydraulic transient processes.

### 4.2. Sensitivity Analysis of Hydro-Mechanical-Electrical System Parameters

To systematically investigate the mechanisms influencing the hydraulic disturbance transient process of pumped-storage power stations under grid-connected operating conditions, this section conducts a comparative study using representative parameters from the hydraulic, mechanical, and electrical domains. Specifically, variations in headrace tunnel length are examined on the hydraulic side, the GVO closing rate of the tripped unit is analyzed on the mechanical side, and the effect of grid inertia is evaluated on the electrical side. The influence mechanisms and corresponding response characteristics associated with each parameter are analyzed and discussed in detail in the subsequent subsections.

#### 4.2.1. Influence of Power System Inertia

In general, a power system with higher inertia possesses a stronger ability to withstand disturbances and operational risks. When a pumped-storage unit is coupled to the grid, the overall grid inertia may influence the dynamic response of the operating unit under hydraulic disturbance conditions. However, the extent of this influence remains unclear. Therefore, this section performs a comparative analysis of hydraulic disturbance transient processes under different grid inertia conditions. The simulation results are presented in Figure 5, and the corresponding extreme values of the hydraulic transient variables are summarized in Table 1.

As shown in Figure 5, the magnitude of grid inertia has only a minor influence on the dynamic response of the operating unit in a pumped-storage power station. Following the load rejection of Unit 1#, both the conventional units participating in primary frequency regulation and the operating unit increase their output to mitigate the frequency drop. In theory, a larger grid inertia results in a smaller frequency decline and correspondingly a smaller reduction in the operating unit’s rotational speed. According to the data in the table, the minimum speed of the operating unit is 0.949 pu when the total system inertia is $H_{sys}$ = 30 s, and 0.961 pu when $H_{sys}$ = 240 s, indicating that higher inertia helps support system frequency. However, the differences remain relatively small.

The main reason is that the range of grid inertia considered is already relatively high, and the units participating in frequency regulation respond quickly, leading to similar frequency trajectories under different inertia values. In addition, the feedback effect of electrical-side frequency disturbances on the hydraulic system is negligible: the maximum volute pressure, minimum draft-tube pressure, and maximum power output remain almost unchanged across different inertia conditions.

In general, the influence of grid inertia on hydraulic feedback can be considered negligible. The dynamic response of the operating unit is dominated primarily by the characteristics of the hydraulic system, with the contribution of active frequency regulation being secondary.

#### 4.2.2. Influence of GVO Closing Time

Because the units in a pumped-storage power station are typically interconnected through bifurcated tunnels, variations in water-hammer pressure can propagate through the hydraulic passages and consequently affect the operating conditions of adjacent units. As a key factor determining the magnitude of water-hammer pressure, the guide vane closing rate of the tripped unit also exerts a significant influence on the dynamic characteristics of the operating unit. Therefore, this section investigates the hydraulic disturbance transient processes under different guide vane closing rates. The simulation results are shown in Figure 6, and the corresponding extreme values of the hydraulic transient variables are summarized in the accompanying Table 2.

As shown in Figure 6, the guide vane closing rate of the tripped unit has a significant influence on the dynamic response of the operating unit. As the closing rate increases, the amplitude of the resulting volute pressure also rises. As illustrated in Figure 6a, when the closing time is 25 s, the maximum spiral-case pressure reaches 480.43 m. The variation in water-hammer pressure further leads to differences in the dynamic characteristics of the operating unit: a higher water-hammer pressure produces a stronger power response. Considering the effect of frequency regulation, the magnitude of the system frequency drop decreases accordingly. When the closing time is 25 s, the minimum system frequency is 0.957 pu, which is higher than that under the other closing rates, as shown in Figure 6b.

Moreover, the change in GVO of the operating unit is mainly driven by the frequency drop on the electrical side, while the influence of hydraulic-pressure fluctuations is comparatively weak. It is worth noting that certain closing patterns may lead to abnormally elevated second or third pressure peaks. This occurs because an excessively slow closing action causes the tripped unit to enter the reverse-“S” region, delaying the onset of flow reversal and thus producing the observed anomaly.

The combined effect of high water-hammer pressure and the increased gate opening induced by frequency regulation results in the maximum power output occurring at a relatively later moment. For instance, when the closing time is 45 s, the maximum power output is 419.81 MW at 41.94 s. Overall, the guide vane closing rate of the tripped unit has a marked impact on dynamic variables such as rotational speed and power output, whereas the gate dynamics of the operating unit remain primarily governed by frequency-induced governor action.

#### 4.2.3. Influence of Tunnel Length

Because the headrace tunnel length directly affects the inertial and elastic characteristics of the hydraulic system, variations in its value lead to differences in the magnitude and propagation behavior of water-hammer pressure. Therefore, this section analyzes the dynamic responses of the operating unit under different headrace tunnel lengths. The corresponding simulation results are presented in Figure 7. and the corresponding extreme values of the hydraulic transient variables are summarized in the Table 3.

As shown in Figure 7, the headrace tunnel length has a pronounced influence on the dynamic response of the operating unit. With increasing tunnel length, the amplitude of the resulting volute pressure also increases. As illustrated in Figure 7a, when the headrace tunnel length is 1500 m, the maximum volute pressure reaches 495.23 m. Changes in the water-hammer pressure further lead to differences in the dynamic characteristics of the operating unit: a larger water-hammer pressure induces a higher power output. Moreover, slight deviations in unit speed can be observed. A longer headrace tunnel results in larger fluctuations in both volute pressure and draft-tube pressure, which cause greater variations in the net head. When the system frequency reaches its minimum, the net head also attains its lowest value; therefore, a longer headrace tunnel corresponds to a lower grid frequency, as shown in Figure 7b.

Similarly, the influence of hydraulic pressure fluctuations on guide vane regulation is relatively weak. The gate opening trajectories are nearly identical for different tunnel lengths, indicating that the gate dynamics are mainly governed by frequency-induced governor action rather than hydraulic pressure variations. The maximum unit output remains closely related to the peak water-hammer pressure. When the headrace tunnel length is 1500 m, the maximum output of the operating unit reaches 432.17 MW. The headrace tunnel length exerts a notable impact on dynamic variables such as unit speed and power output, whereas the guide vane dynamics remain predominantly driven by frequency regulation triggered by the transient frequency drop.

Overall, a lower grid inertia leads to a larger decline in grid frequency, while its influence on spiral-case pressure can be considered negligible. In contrast, a shorter guide vane closing time and a longer headrace tunnel length result in higher spiral-case pressure, whereas their effects on grid frequency are insignificant. These observations indicate that the pump–turbine plays a partial decoupling role between the hydraulic system and the power grid, causing the influence of different parameters to be predominantly manifested within their respective physical domains. However, it should be emphasized that the pump–turbine can only mitigate, rather than completely eliminate, the interactions between the hydraulic and electrical subsystems. Therefore, the hydro-electrical coupling mechanisms between pumped-storage units and the power grid require further clarification and in-depth investigation.

## 5. Discussion

Compared with previous studies, incorporating the dynamic characteristics of the power system introduces new coupling relationships among pumped-storage units, indicating that further research is required in this area. Future investigations may focus on the following aspects:**(1)** **Additional influencing factors**

In multi-unit pumped-storage power stations, unit interactions are not limited to the hydraulic domain but also arise through electrical interconnections, which can significantly influence the dynamic regulation behavior of individual units. As a result, units traditionally regarded as independent in simplified analyses may exhibit implicit coupling through the electrical subsystem. The governor, as the core control component, plays a critical role in shaping transient system responses, and different control modes can substantially affect unit dynamics and the associated hydro-electrical coupling characteristics. In addition, several important factors, including governor parameter settings, water conveyance system configuration, system scale, and grid strength, may further influence the dynamic security of operating units. A systematic investigation of these aspects is therefore necessary to achieve a more comprehensive understanding of coupled hydro-electro-mechanical dynamics under various operating conditions.

**(2)** 
**Mechanism Analysis and Contribution Assessment**


Based on the sensitivity analysis presented in Section 4, it is observed that parameters such as grid inertia, guide vane closing time, and the length of the headrace tunnel exert noticeable influences on the dynamic characteristics of the system. These results indicate that, under grid-connected operating conditions, the system exhibits pronounced multi-physical coupling among the hydraulic, mechanical, and electrical subsystems, highlighting the necessity of considering hydro-electro-mechanical interactions in the analysis. However, it should also be emphasized that different system variables affect the dynamic response and extreme values of the system to varying degrees. Quantitatively characterizing the coupling strength and contribution of these parameters is essential for achieving a deeper understanding of the interaction mechanisms among different physical domains. Such a contribution-based coupling analysis represents an important direction for future research and constitutes a key focus of our ongoing work.

## 6. Conclusions

This study develops an integrated hydraulic–mechanical–electrical grid coupled transient model for pumped-storage power stations and systematically investigates the hydraulic disturbance process under detailed grid dynamic characteristics. The proposed model is validated using data from a physical model test platform. By comparing a detailed grid model with a conventional simplified representation, the influence mechanism of electrical coupling on hydraulic transients is revealed. In addition, key parameters from the hydraulic, mechanical, and electrical domains are examined to analyze their effects on unit dynamic responses. The major conclusions are as follows:(1)Detailed grid modeling has a significant impact on the hydraulic disturbance transient process of pumped-storage power stations. When grid dynamics are considered, load rejection causes a frequency drop, and the operating unit must compensate for the resulting power deficit. This leads to a decrease in its rotational speed and an increase in GVO, which helps mitigate water-hammer pressure to some extent. The unit rotational speed and guide vane opening exhibit response trends that differ from those predicted by simplified grid models. This phenomenon is associated with grid-unit interaction mechanisms and is not limited to a specific water conveyance layout.(2)Hydraulic–mechanical parameters, such as the length of the headrace tunnel and the guide vane closing rate, have significant impacts on the dynamic characteristics of hydraulic disturbance. These parameters directly affect the magnitude of water-hammer pressure and the peak power output. Meanwhile, the dynamic evolution of guide vane opening is strongly influenced by grid-frequency regulation and control strategies, indicating that the observed dynamic behaviors are closely related to the adopted governor control mode.(3)Variations in grid inertia have relatively minor effects on the hydraulic disturbance process within the range investigated in this study. Within the scope of this study, differences in inertia lead to only small variations in unit rotational speed, pressure, and power responses. The dynamic behavior of the operating unit is still dominated by the characteristics of the hydraulic system.The influence of grid inertia may become more pronounced under other grid conditions or control strategies and warrants further investigation.

Overall, these results demonstrate that electrical coupling plays a critical role in shaping hydraulic disturbance dynamics and must be considered for accurate modeling. Meanwhile, hydraulic–mechanical parameters remain dominant in determining the system’s transient severity, whereas grid inertia contributes comparatively limited influence. The findings highlight the importance of integrated hydraulic–mechanical–electrical grid modeling for evaluating the dynamic security of pumped-storage power stations under disturbance conditions.

## Figures and Tables

**Figure 1 sensors-26-00311-f001:**
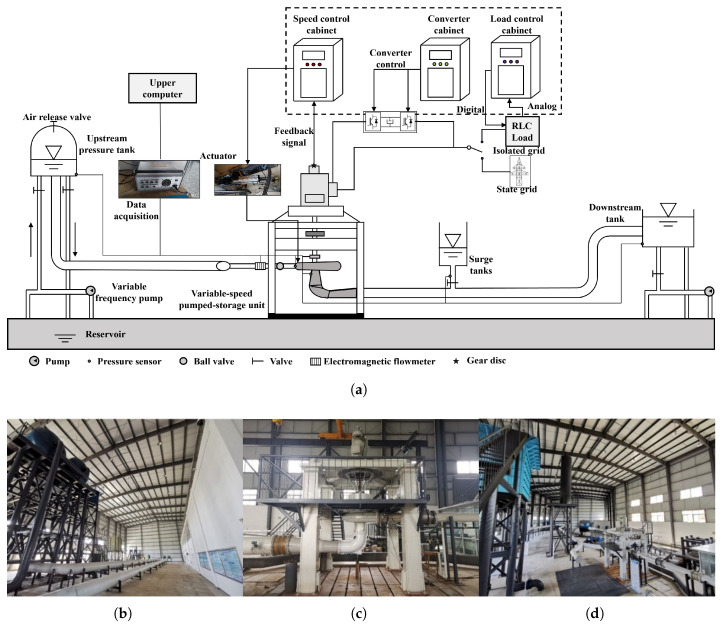
Physical model test platform of the pumped-storage power station. (**a**) Schematic diagram of the model test platform. (**b**) Photograph of upstream side. (**c**) Photograph of pump-turbine unit. (**d**) Photograph of downstream side.

**Figure 2 sensors-26-00311-f002:**
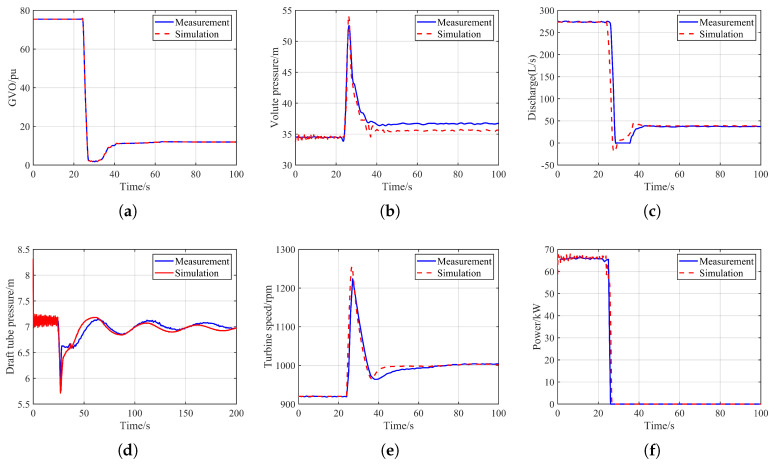
Comparison Between Experimental and Simulation Results. (**a**) GVO. (**b**) Volute pressure. (**c**) Unit flow. (**d**) Draft tube pressure. (**e**) Turbine speed. (**f**) Output power.

**Figure 3 sensors-26-00311-f003:**
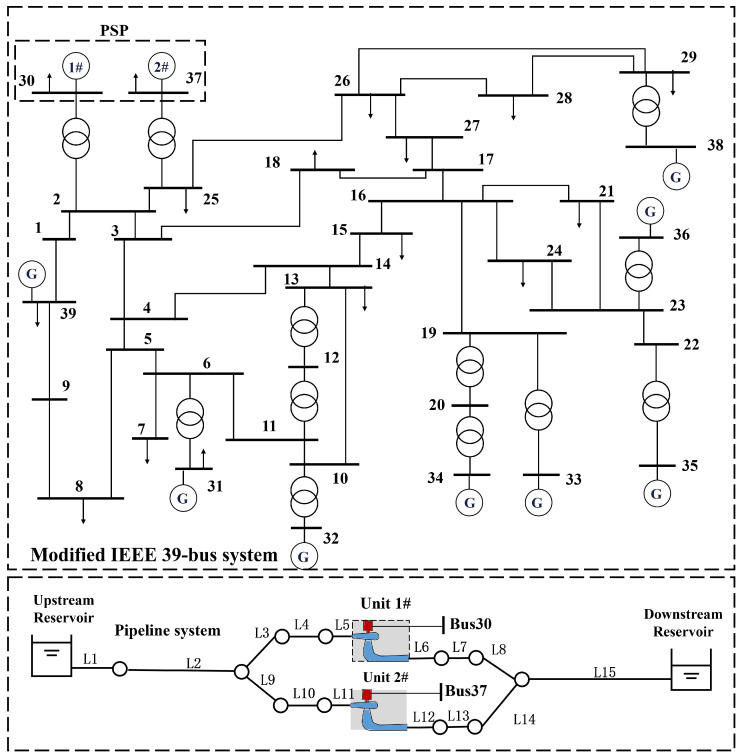
Modified IEEE 39-bus system coupled with the hydraulic system of PSP.

**Figure 4 sensors-26-00311-f004:**
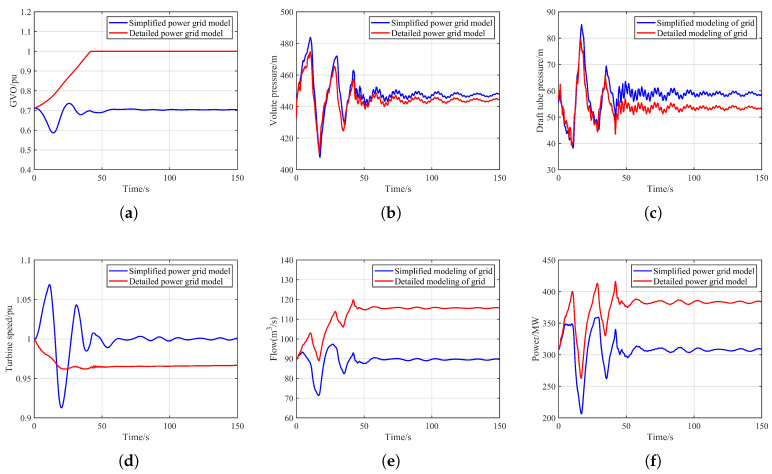
Comparison of simulation results for normal unit hydraulic transition process under different grid model. (**a**) GVO. (**b**) Volute pressure. (**c**) Draft tube pressure. (**d**) Turbine speed. (**e**) Unit flow. (**f**) Output power.

**Figure 5 sensors-26-00311-f005:**
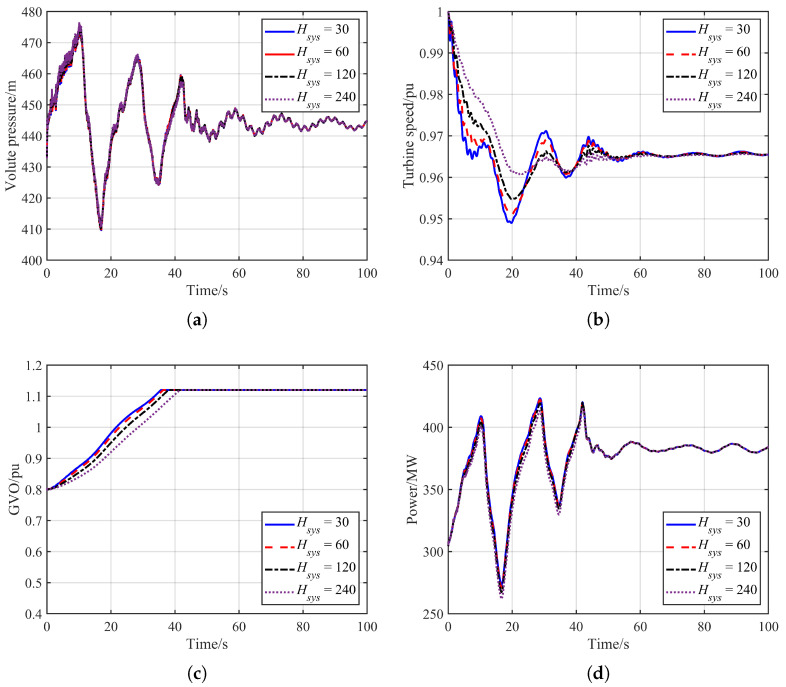
Comparison of simulation results under different grid inertia levels. (**a**) Volute pressure. (**b**) Turbine speed. (**c**) GVO. (**d**) Unit power output.

**Figure 6 sensors-26-00311-f006:**
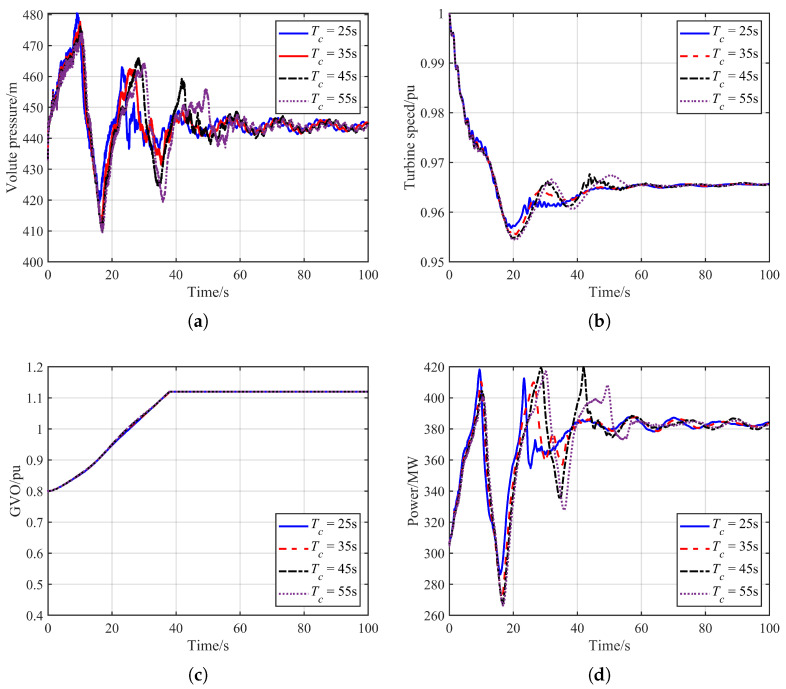
Comparison of simulation results under different GVO closing Time. (**a**) Volute pressure. (**b**) Turbine speed. (**c**) GVO. (**d**) Unit power output.

**Figure 7 sensors-26-00311-f007:**
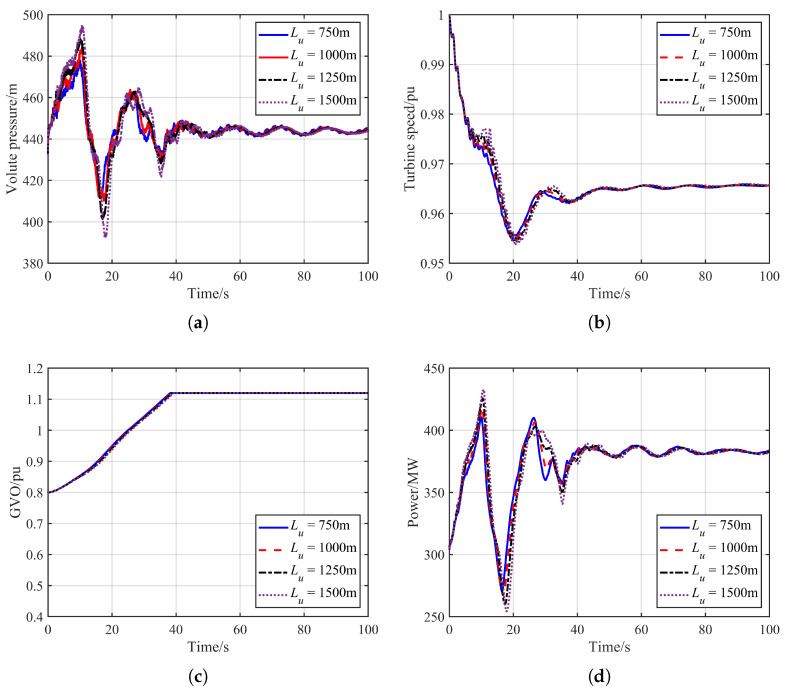
Comparison of simulation results under differentheadrace tunnel length. (**a**) Volute pressure. (**b**) Turbine speed. (**c**) GVO. (**d**) Unit power output.

**Table 1 sensors-26-00311-t001:** Extreme values of hydraulic transient variables under different grid inertia levels.

Grid Inertia	Maximum Volute Pressure/m	Minimum Draft-Tube Pressure/m	Minimum Unit Speed/pu	Maximum Unit Output/MW
$H_{sys}$ = 30	475.16	38.83	0.949	423.35
$H_{sys}$ = 60	475.64	38.82	0.951	422.15
$H_{sys}$ = 120	476.16	38.89	0.954	419.81
$H_{sys}$ = 240	476.62	38.91	0.961	415.91

**Table 2 sensors-26-00311-t002:** Extreme values of hydraulic transient variables under different GVO closing time.

Grid Inertia	Maximum Volute Pressure/m	Minimum Draft-Tube Pressure/m	Minimum Unit Speed/pu	Maximum Unit Output/MW
$T_{c}$ = 25 s	480.43	33.83	0.957	418.30
$T_{c}$ = 35 s	477.69	36.97	0.955	410.31
$T_{c}$ = 45 s	476.15	38.89	0.954	419.81
$T_{c}$ = 55 s	474.32	39.64	0.954	417.11

**Table 3 sensors-26-00311-t003:** Extreme values of hydraulic transient variables under different Headrace tunnel length.

Grid Inertia	Maximum Volute Pressure/m	Minimum Draft-Tube Pressure/m	Minimum Unit Speed/pu	Maximum Unit Output/MW
$L_{u}$ = 750 m	477.85	36.91	0.955	410.52
$L_{u}$ = 1000 m	483.50	37.72	0.954	417.59
$L_{u}$ = 1250 m	488.15	38.14	0.954	425.69
$L_{u}$ = 1500 m	495.23	39.47	0.953	432.17

## Data Availability

The original contributions presented in this study are included in the article. Further inquiries can be directed to the corresponding authors.

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
