# Peer review of "Study of Hydraulic Disturbance Transient Processes in Pumped-Storage Power Stations Considering Electro-Mechanical Coupling"

_sensors, 2026, doi:10.3390/s26010311_

Round 1
Reviewer 1 Report
Comments and Suggestions for Authors
This manuscript presents a well-organized and thorough study on hydraulic disturbance transient processes in pumped-storage power stations with detailed electro-mechanical coupling. The integration of a full power-system model with hydraulic–mechanical dynamics is valuable, and the experimental validation strengthens the credibility of the results. Only minor improvements are suggested to further enhance clarity and readability.
- The contributions are clear; adding one or two sentences in the Introduction to more explicitly highlight how the integrated model improves over conventional approaches would make the novelty stand out even more.
- The manuscript is generally very well written. A light proofreading to unify terminology (e.g., consistent naming of “guide vane opening,” “GVO,” etc.) and ensure consistent unit formatting would further polish the presentation.
- The model of the power system is still too complex. Are there any other models that can be used as a substitute?
- Does the coupling interference between the power system and the hydraulic system only manifest itself during the transition process of hydraulic interference?
Author Response
This manuscript presents a well-organized and thorough study on hydraulic disturbance transient processes in pumped-storage power stations with detailed electro-mechanical coupling. The integration of a full power-system model with hydraulic–mechanical dynamics is valuable, and the experimental validation strengthens the credibility of the results. Only minor improvements are suggested to further enhance clarity and readability.
Comments 1: The contributions are clear; adding one or two sentences in the Introduction to more explicitly highlight how the integrated model improves over conventional approaches would make the novelty stand out even more.
Response 1: Thank you for your suggestion and comments. Although a discussion on the limitations of traditional models has been added in the Introduction, the original description was not sufficiently accurate. The revised description is provided as follows:
However, these studies usually involve substantial simplifications of the electrical subsystem, typically adopting single-machine isolated-grid models or infinite-bus representations. Such modeling approaches result in an overly simplified description of the power system, especially in the context of modern power system structures with increasing complexity and high penetration of renewable energy sources. As a result, these simplified models are generally unable to capture the electrical coupling effects between the power grid and the generating units, thereby limiting their ability to reflect the coupled dynamic characteristics of the pumped storage units under realistic operating conditions. (Page 2, Line 96).
In addition, following the reviewers’ comments and suggestions, the description of the contributions of this work has been revised to explicitly highlight the improvements of the adopted model over traditional modeling approaches.
A mathematical model of the transient processes in pumped-storage power stations is developed, incorporating a detailed representation of grid dynamics that captures variations in both voltage and frequency, as well as the dynamic characteristics of individual system components. The accuracy of the proposed model is validated using data obtained from a physical model test platform. (Page 2, Line 108).
Comments 2: The manuscript is generally very well written. A light proofreading to unify terminology (e.g., consistent naming of “guide vane opening,” “GVO,” etc.) and ensure consistent unit formatting would further polish the presentation.
Response 2: Thank you for your suggestion and comments. A careful proofreading has been conducted throughout the manuscript to unify terminology, including the consistent use and definition of terms such as “guide vane opening (GVO),” and to ensure uniform formatting of all units.
Comments 3: The model of the power system is still too complex. Are there any other models that can be used as a substitute?
Response 3: Thank you for your suggestion and comments. The power system model adopted in this study is designed to accurately capture the electromechanical interactions between the power grid and the pumped storage power station. Although a variety of simplified models, such as single-machine isolated-grid models and reduced-order equivalent network models, can describe the relationship between grid frequency and unit dynamics to some extent, the coupling between the power grid and pumped storage plants is inherently multi-dimensional.
In particular, critical interactions involving voltage variations, grid frequency dynamics, and their successive and mutual influences cannot be sufficiently represented by simplified models. Moreover, with the increasing complexity of modern power systems, reduced mathematical models are often inadequate to fully reflect the practical operating conditions and challenges faced by pumped storage power stations.
For these reasons, a relatively detailed power system model is employed in this work to comprehensively investigate the interactions between the power grid and the pumped storage plant, thereby enhancing the fidelity and engineering relevance of the obtained results.
Comments 4: Does the coupling interference between the power system and the hydraulic system only manifest itself during the transition process of hydraulic interference?
Response 4: Thank you for your suggestion and comments. Under grid-connected operating conditions, the coupling between the hydraulic system and the power system is ubiquitous and does not exist solely during hydraulic disturbance processes. In the specific case of hydraulic disturbances, the hydraulic transmission mechanism indeed plays a dominant role; however, the coupling influence of the electrical system cannot be neglected. In particular, variations in grid frequency can directly affect the governor response, altering its control action and subsequently inducing dynamic changes in the hydraulic system. This interaction is already evident even under hydraulic disturbance scenarios. Moreover, during normal operating conditions, fluctuations in electrical load, voltage variations, and frequency deviations in the power system are continuously transmitted to the pumped storage plant, thereby influencing its operating state and performance. These coupling effects become more pronounced when pumped storage units operate under asymmetric conditions or are connected to different power grids, where electrical–hydraulic interactions may be further amplified.
In addition, the hydraulic–electrical coupling is not confined to transient hydraulic processes, although transient events tend to amplify the coupling effects and make them more observable. Under steady-state operating conditions, hydraulic variables (such as flow rate and pressure) and electrical variables (such as output current and electromagnetic torque) are mutually constrained. Even in the absence of abrupt disturbances, any change in electrical operating conditions may lead to a redistribution of variables in the water conveyance system, and vice versa. This bidirectional interaction reflects the inherent coupling nature of the hydraulic and power systems, which persists in both steady-state and transient regimes, with transient processes primarily acting as an amplification mechanism. Therefore, it would be overly restrictive to attribute the coupling effects solely to the hydraulic system, as the power system also plays a critical and active role.
Reviewer 2 Report
Comments and Suggestions for Authors
The article presented by the authors is topical, but the following minor corrections are necessary:
- The terms of the relation (4) should be explained.
- At row R 159, the explanation of the terms of the relation (5) should be written more carefully.
- In Figure 2, the explanations (from c. to e.) do not correspond to the figures. It should be corrected.
- Figures 2 and 4 have the same name. They should be customized. The same goes for Figures 5, 6 and 7.
- For chapter 5 "Discussions" I recommend the authors to discuss the results obtained in chapter 4.
Author Response
The article presented by the authors is topical, but the following minor corrections are necessary:
Comments 1: The terms of the relation (4) should be explained.
Response 1: Thank you for your suggestion and comments. The physical meanings of all terms in Eq. (4) have now been explicitly explained in the revised manuscript. Additional descriptions have been added immediately after Eq. (4) to clarify the definition and role of each term.
where, CQP​, CQM ​, QCP, and QCM are intermediate variables calculated from Eq. (3). The coefficients Ct ​, Ct1 ​, Ct2 ​, Ct3 ​, Ct4 ​, and Ct5 are auxiliary intermediate variables introduced for the computation of CQP​, CQM ​, QCP, and QCM. (Page 3, Line 144).
Comments 2: At row R 159, the explanation of the terms of the relation (5) should be written more carefully.
Response 2: Thank you for your suggestion and comments. The explanation of the terms in Eq. (5) has been carefully revised at row R159. The descriptions have been rewritten to more clearly and precisely explain the physical meaning and role of each term. The revised explanation is as follows:
where Q is the discharge, m3/s; N is the rotational speed, r/min; M is the turbine torque, N·m; H is the pump-turbine working head, m; D1 is the runner diameter, m; Q11 is the unit discharge, m3/s; N11 is the unit speed, r/min; and M11 is the unit torque, N·m. (Page 4, Line 144).
Comments 3: In Figure 2, the explanations (from c. to e.) do not correspond to the figures. It should be corrected.
Response 3: Thank you for your suggestion and comments. The subfigure explanations in Fig. 2 (c)–(e) have been carefully checked and corrected to ensure they correspond to the correct subfigures. The labels and captions have been updated accordingly in the revised manuscript.
Comments 4: Figures 2 and 4 have the same name. They should be customized. The same goes for Figures 5, 6 and 7.
Response 4:Thank you for your suggestion and comments. The titles of Figs. 2 and 4 have been revised to ensure they are distinct and appropriately customized. The same corrections have also been applied to Figs. 5, 6, and 7, and all figure titles have now been checked to avoid duplication and improve clarity.
Comments 5: For chapter 5 "Discussions" I recommend the authors to discuss the results obtained in chapter 4.
Response 5:Thank you for your suggestion and comments. Following your suggestion, a summarizing discussion has been added at the end of Chapter 4. In addition, the Discussion section has been further refined and revised to improve clarity and coherence. The added summary for Chapter 4 is presented as follows:
Overall, a lower grid inertia leads to a larger decline in grid frequency, while its influence on spiral-case pressure can be considered negligible. In contrast, a shorter guide vane closing time and a longer headrace tunnel length result in higher spiral-case pressure, whereas their effects on grid frequency are insignificant. These observations indicate that the pump–turbine plays a partial decoupling role between the hydraulic system and the power grid, causing the influence of different parameters to be predominantly manifested within their respective physical domains. However, it should be emphasized that the pump–turbine can only mitigate, rather than completely eliminate, the interactions between the hydraulic and electrical subsystems. Therefore, the hydro-electrical coupling mechanisms between pumped-storage units and the power grid require further clarification and in-depth investigation. (Page 17, Line 453).
The revised Discussion is presented as follows:
(1)Additional influencing factors
In multi-unit pumped-storage power stations, unit interactions are not limited to the hydraulic domain but also arise through electrical interconnections, which can significantly influence the dynamic regulation behavior of individual units. As a result, units traditionally regarded as independent in simplified analyses may exhibit implicit coupling through the electrical subsystem. The governor, as the core control component, plays a critical role in shaping transient system responses, and different control modes can substantially affect unit dynamics and the associated hydro--electrical coupling characteristics. In addition, several important factors, including governor parameter settings, water conveyance system configuration, system scale, and grid strength, may further influence the dynamic security of operating units. A systematic investigation of these aspects is therefore necessary to achieve a more comprehensive understanding of coupled hydro-electro-mechanical dynamics under various operating conditions.
(2)Mechanism Analysis and Contribution Assessment
Based on the sensitivity analysis presented in Section 4, it is observed that parameters such as grid inertia, guide vane closing time, and the length of the headrace tunnel exert noticeable influences on the dynamic characteristics of the system. These results indicate that, under grid-connected operating conditions, the system exhibits pronounced multi-physical coupling among the hydraulic, mechanical, and electrical subsystems, highlighting the necessity of considering hydro-electro-mechanical interactions in the analysis. However, it should also be emphasized that different system variables affect the dynamic response and extreme values of the system to varying degrees. Quantitatively characterizing the coupling strength and contribution of these parameters is essential for achieving a deeper understanding of the interaction mechanisms among different physical domains. Such a contribution-based coupling analysis represents an important direction for future research and constitutes a key focus of our ongoing work. (Page 17, Line 469).
Reviewer 3 Report
Comments and Suggestions for Authors
The scientific article "Study of Hydraulic Disturbance Transient Processes in
Pumped-Storage Power Stations Considering Electro-mechanical Coupling" is devoted to the development of a coupled model of a grid-connected pumped-storage power station incorporating a detailed representation of the power system. The paper is well-written, and the quality of the figures is acceptable. The paper is attractive and has some valuable conclusions. The following are my observations in this regard.
1. The conclusions of the paper need to be generalized further. Do they apply only to the configuration of a pumped-storage station (single-tunnel–two-unit)? It is necessary to describe the extent to which the conclusions of the work apply to other configurations.
2. The hydraulic model does not take into account the compressibility of water and the elasticity of pipes. This should be described, and its impact explained.
3. How will the results of the work be used in the design and configuration of the actual configuration?
In general, I think the manuscript can be accepted with minor changes.
Author Response
The scientific article "Study of Hydraulic Disturbance Transient Processes in Pumped-Storage Power Stations Considering Electro-mechanical Coupling" is devoted to the development of a coupled model of a grid-connected pumped-storage power station incorporating a detailed representation of the power system. The paper is well-written, and the quality of the figures is acceptable. The paper is attractive and has some valuable conclusions. The following are my observations in this regard.
Comments1: The conclusions of the paper need to be generalized further. Do they apply only to the configuration of a pumped-storage station (single-tunnel–two-unit)? It is necessary to describe the extent to which the conclusions of the work apply to other configurations.
Response 1: Thank you for your suggestion and comments. The single-tunnel–two-unit configuration considered in this study represents one typical layout of pumped-storage power stations, rather than a limitation of the proposed modeling approach. When a detailed grid model is taken into account, the dynamic characteristic that grid frequency decreases following load rejection is a general phenomenon, which is independent of the specific layout of the water conveyance system adopted by the pumped-storage station. Moreover, the dynamic response of pumped-storage units is primarily determined by the applied control strategy. Under the same regulation mode considered in this work, the dynamic behavior of the operating units remains consistent with the mechanisms described in the conclusions . Nevertheless, we agree that some of the original conclusions were stated in an overly absolute manner. Therefore, the conclusions have been carefully revised to improve their generality and to clarify the extent to which they apply to different pumped-storage configurations and operating scenarios. The revised conclusions are presented as follows:
(1) Incorporating detailed grid modeling has a significant impact on the hydraulic disturbance transient process of pumped-storage power stations. When grid dynamics are considered, load rejection causes a frequency drop, and the operating unit must compensate for the resulting power deficit. This leads to a decrease in its rotational speed and an increase in guide vane opening, which helps mitigate water-hammer pressure to some extent. The unit rotational speed and guide vane opening exhibit response trends that differ from those predicted by simplified grid models. This phenomenon is associated with grid-unit interaction mechanisms and is not limited to a specific water conveyance layout.
(2) Hydraulic-mechanical parameters, such as the length of the headrace tunnel and the guide vane closing rate, play a dominant role on the dynamic characteristics of hydraulic disturbance. These parameters directly affect the magnitude of water-hammer pressure and the peak power output. Meanwhile, the dynamic evolution of guide vane opening is strongly influenced by grid-frequency regulation and control strategies, indicating that the observed dynamic behaviors are closely related to the adopted governor control mode.
(3) Variations in grid inertia exhibit a relatively limited influence on the hydraulic disturbance process within the range investigated in this study. Within the scope of this study, differences in inertia lead to only small variations in unit rotational speed, pressure, and power responses. The dynamic behavior of the operating unit is still dominated by the characteristics of the hydraulic system. The influence of grid inertia may become more pronounced under other grid conditions or control strategies and warrants further investigation. (Page 18, Line 504)
Comments 2: The hydraulic model does not take into account the compressibility of water and the elasticity of pipes. This should be described, and its impact explained.
Response 2: Thank you for your suggestion and comments. In this study, the hydraulic transient process is modeled using the method of characteristics (MOC), in which the effects of water compressibility and pipe-wall elasticity are implicitly represented by an equivalent wave speed parameter. Although the wave speed may theoretically vary when water compressibility and pipe elasticity are modeled explicitly, such variations are generally limited for the water conveyance systems of pumped-storage power stations and can be reasonably approximated as constant during transient events. This assumption is widely adopted in practical hydraulic transient analysis. In this work, a constant wave speed is assumed mainly to improve computational efficiency while maintaining sufficient accuracy for capturing the essential features of hydraulic transients and their coupling with the electrical system. The possible influence of wave speed variation is therefore expected to affect only quantitative details, rather than the qualitative conclusions of this study.
Comments 3: How will the results of the work be used in the design and configuration of the actual configuration?
Response 3: Thank you for your suggestion and comments. The results of this study can be applied to practical engineering from multiple perspectives.
From a design perspective, during the design stage of pumped-storage power stations, stability assessment is traditionally conducted using simplified models, which are feasible but may lead to inaccurate evaluations and potentially uneconomic system configurations. The proposed approach provides a more accurate basis for stability analysis and can help improve design decisions.
From an operational and regulation perspective, incorporating a detailed power system model allows the analysis to better reflect realistic grid conditions. This facilitates the effective utilization of the regulation capability of pumped-storage power stations while ensuring secure and stable operation under modern power system environments.
In general, I think the manuscript can be accepted with minor changes.
Response: Thank you for your suggestion and comments. We have revised the paper according to your suggestions.